# Exercise Referral Instructors’ Perspectives on Supporting and Motivating Participants to Uptake, Attend and Adhere to Exercise Prescription: A Qualitative Study

**DOI:** 10.3390/ijerph19010203

**Published:** 2021-12-25

**Authors:** Colin B. Shore, Stuart D. R. Galloway, Trish Gorely, Angus M. Hunter, Gill Hubbard

**Affiliations:** 1Faculty of Health and Medical Sciences, School of Health Sciences, University of Surrey, Guildford GU2 7YH, UK; 2Physiology, Exercise and Nutrition Research Group, Faculty of Health Sciences and Sport, University of Stirling, Stirling FK9 4LA, UK; s.d.r.galloway@stir.ac.uk (S.D.R.G.); angus.hunter@ntu.ac.uk (A.M.H.); 3Department of Nursing and Midwifery, University of the Highlands and Islands (UHI), Inverness IV2 3JH, UK; trish.gorely@uhi.ac.uk (T.G.); gill.hubbard@uhi.ac.uk (G.H.); 4Department of Sports Science, Nottingham Trent University, Nottingham NG11 8NS, UK

**Keywords:** community-based research, exercise prescription, physical activity, motivation, public health practice, behaviour change

## Abstract

Exercise referral schemes are designed to support people with non-communicable diseases to increase their levels of exercise to improve health. However, uptake and attendance are low. This exploratory qualitative study aims to understand uptake and attendance from the perspectives of exercise referral instructors using semi-structured interviews. Six exercise referral instructors from one exercise referral scheme across four exercise referral sites were interviewed. Four themes emerged: (i) the role that instructors perceive they have and approaches instructors take to motivate participants to take-up, attend exercise referral and adhere to their exercise prescription; (ii) instructors’ use of different techniques, which could help elicit behaviour change; (iii) instructors’ perceptions of participants’ views of exercise referral schemes; and (iv) barriers towards providing an exercise referral scheme. Exercise referral instructors play an important, multifaceted role in the uptake, attendance and adherence to exercise referral. On-going education and peer support for instructors may be useful. Instructors’ perspectives help us to further understand how health and leisure services can design successful exercise referral schemes.

## 1. Introduction

Physical inactivity has been shown to increase the risk of developing many non-communicable diseases (NCD), including diabetes, certain types of cancer and heart disease and is currently one of the fourth leading risk factors for mortality [1]. NCDs are caused by and share, to a large extent, four behavioural risk factors; tobacco use, unhealthy diet, physical inactivity and the harmful use of alcohol, all of which are common facets of 21st-century lifestyles [1]. Physical inactivity (6%) and tobacco use (9%) have been identified as two of the four leading risk factors contributing towards global mortality (6% of deaths globally) alongside high blood pressure (13%) and high blood glucose (6%) [1]. A wide range of approaches has been employed to increase physical activity (PA) levels, both at the population and individual level. One of those approaches, in the United Kingdom, is exercise referral schemes (ERS). ERS are complex, multifaceted interventions; often multi-agency between local general medical practices or health services and voluntary, local council or privately-operated leisure service providers [2]. ERS involves the referral of individuals deemed to be insufficiently inactive and or suffering from, or at risk of developing NCDs, from health practitioners (e.g., general practitioner, nurse) to an exercise programme [3]. ERS participants undertake programmes of supervised, safe, appropriate, prescribed exercises, where they may benefit physiologically and psychologically from increasing their level of PA [2]. However, the evidence base for ERS has been questioned, notably the inconsistent and weak evidence base and assessment [4,5], limited increase in PA levels [6,7], wellbeing, quality of life or health outcomes [7,8] and inequalities [9]. 

Any physiological and psychological benefits that may be achieved as an outcome of ERS is initially reliant on participants’ uptake and attendance at the scheme and adherence to the prescribed exercise. In turn, uptake (attending the first ERS session), attendance (continued presence at ERS, measured by number of sessions completed) and adherence (completion of prescribed exercise and continued attendance) are in part reliant on an exercise referral instructor’s ability to motivate participants to uptake, attend and adhere to the prescribed prescription. It is important to acknowledge that additional variables such as gender, socioeconomic status or medical condition also influence uptake, attendance and adherence. However, the evidence supporting this is mixed [4] and is not the primary focus of this study. Therefore, there is a need for a good relationship between the participant and the exercise referral instructor. However, relationships between instructors and participants are rarely simple. Relationships can encounter challenges and fluctuations according to balances between personal objectives, oganisational objectives and the cooperation and reflexivity inherent within the relationship [10,11].

To date, most research has focused on attributes of participants’ value in their instructors. For example, access to an instructor who is knowledgeable and effective at providing interpersonal support is a key determinant of increased attendance within ERS [12,13,14]. A mixed-methods process evaluation reported that comprehensive professional contact (from instructors) provided motivational support that enabled participants to build confidence in becoming more active from a practical standpoint. That is, they felt confident in performing the exercise [15]. Previous work has shown that successful instructors are able to create, represent, advance and embed a shared sense of identity among group members [16]. This is suggestive that instructors are role models and can influence (both positively and negatively) members’ exercise experiences and intentions to continue engaging in ERS [17,18].

However, there is a paucity of evidence surrounding experiences of exercise referral instructors and how they engage, support and motivate participants to uptake and attend the scheme and adhere to the prescription. Historically, ERSs have a low uptake rate, and for those who choose to start ERS, there is a high dropout rate or low session count, i.e., attendance is poor [7,19]. For example, uptake can range anywhere between 35 and 85%, while attendance can range from 12 to 86% [4]. Therefore, it is key to try to understand exercise referral instructors’ influence by capturing current practice and approaches undertaken to ensure participants’ uptake, attendance and adherence to the prescription. Such insight may further our understanding of how ERS can play a role in primary and secondary prevention and self-management of NCDs. Therefore, the aim of the study was to explore exercise referral instructors’ perceptions and experiences of motivating and supporting participants to uptake and attend ERS and ensure adherence to the prescribed exercise. 

## 2. Materials and Methods

### 2.1. Study Design

An exploratory qualitative study involving descriptive thematic analysis of semi-structured interviews was conducted with exercise referral instructors. The University of Stirling NHS, Invasive or Clinical Research Panel granted Ethics’ approval (NICR (17/18) Paper No.004, October 2017). 

### 2.2. Exercise Referral Scheme

The ERS operated in four different leisure centres in Western Scotland. The four different leisure centres were under the administrative control of the local council. Each individual leisure centre also provided exercise classes and facilities open to the general public. The ERS accepted referrals from local healthcare professionals (HCPs) of adults aged 18 years or above who were deemed to not be meeting PA guidelines (at least 150 min (2 1/2 h) of moderate-intensity activity or 75 min of vigorous-intensity activity, performing activities to develop or maintain strength in the major muscle groups on at least two days a week and reducing the amount of time spent being sedentary [20]), and/or suffering from a medical condition that could potentially benefit from an exercise programme. Each exercise referral site operated two referral exercise sessions per week, held on Tuesdays and Thursdays at the cost of GBP 3.90 per session. The duration of the exercise programme was 12-weeks. Referred participants were given a paper referral with the details of the ERS. Participants were responsible for contacting their local leisure centre, either by telephone or presenting themselves at the leisure centre with their paper referral from the HCPs, in order to arrange an initial appointment with an exercise referral instructor. During the initial appointment, an introductory interview was conducted by the exercise referral instructor, who would also be the instructor that prescribed their exercise and led the programme. Participants were made aware of the exercise programme, timings of sessions and price, and given a tour of the leisure centre facility; exercise referral instructors also reviewed the participants’ referral medical condition. Instructors also established any potential barriers or facilitators to undertaking the programme, including any goals from participation in the exercise programme. No exercise was completed in the initial appointment. The exercise session was held in a gymnasium setting, within the leisure centre, making use of resistance machines, free weights or using bodyweight as a means of resistance and cardiovascular machines. Broadly, each exercise session was 1-hr long and consisted of an aerobic warm-up, followed by a selection of aerobic exercises completed on a variety of cardiovascular machines. Resistance exercises were completed afterwards. A detailed breakdown of the frequency, intensity, type and time of each exercise prescribed by the participants ERS is available [21]. Exercise referral instructors provided each participant with a personalised prescription card and encouraged them to perform the exercises independently whilst being closely monitored by the instructor. One instructor would lead the session, with an upper limit of 10 participants per instructor. All participants exercise independently, but all ERS participants attended at the same time and were encouraged to interact, but it was not a group class. After completion of the ERS, participants were offered the chance to become full paying members of the leisure centre. It is important to recap the terms, uptake, attendance and adherence. Uptake is understood as the participant attending their first exercise referral session. Attendance is defined as continued presence at the ERS and is measured by the number of sessions completed. Adherence relates to the prescribed exercise programme and can be recorded as a combination of session count (attendance) and performing the required exercise prescription. Exercise referral instructors were employed either part- or full-time. In order to be employed as an exercise referral instructor, individuals had to hold a minimum level three diploma in Fitness Instructing and Personal Training and possess a level three exercise referral qualification to lead exercise referral sessions [22]. 

### 2.3. Recruitment and Participant Inclusion

A purposive, informal email approach was made to a Scottish ERS enquiring about interest in taking part in the study. Upon showing interest in the study, one author (Colin B. Shore) attended a meeting with all exercise referral instructors and an ERS manager to outline the study. Participant information sheets were distributed to instructors with time allocated for instructors to ask questions. Further contact details were made available to instructors if they had further questions. Instructors were then invited to participate in the study; if they agreed, a suitable time and date were set for one author (Colin B. Shore) to visit the referral sites to conduct the one-on-one interviews. Inclusion criteria for study participants were any instructor aged 18 or above who possessed the appropriate qualifications (described above) and delivered exercise referral sessions. Before commencement of interviews, each ERS instructor was provided with a brief verbal recap of the purpose and format of the interview, alongside assurances of confidentiality and a further opportunity to withdraw if required. 

### 2.4. Qualitative Data Collection

Given the exploratory and inductive focus of the research on exercise professionals’ perceptions of ERS, face-to-face, semi-structured interviews were undertaken. Interviews were conducted at ERS sites, where instructors worked in a quiet meeting room. The interviews were guided by a semi-structured schedule. The schedule was developed using a literature search and collaborative discussions among the authors. The nature of the questions broadly covered perceptions of ERS referral, the role of an ERS instructor, including motivational strategies relating to uptake, attendance and adherence of exercise prescription and assessing progress [21]. An initial pilot of the questions was conducted with a qualified exercise instructor, independent of the study, who had experience in prescribing and leading exercise classes for participants with long-term health conditions. The schedule was not designed to have questions posed chronologically, but rather in an order that seemed to follow the natural flow of the conversation. Open-ended questions allowed instructors the opportunity to express their experiences, providing deeper and detailed insights into their experiences. All ERS instructors provided verbal and written informed consent for digital audio (Olympus VN-731PC) recording and use of anonymised quotations.

### 2.5. Qualitative Analysis

Data were transcribed verbatim with transcripts and sound files were stored in an encrypted research drive. A descriptive thematic approach to analysis was used and adapted from Braun and Clarke [23]’s six-phase guide. Five steps are described below, while the sixth step is the write-up of the results [23]. The analysis was approached from a pragmatic epistemic position, guided by the authors’ research experiences rather than a specific theoretical framework. This approach is particularly helpful as it is designed to answer the question of whether an intervention works under usual conditions, rather than explanatory trials that answer the question of whether an intervention works under “ideal conditions” [24,25]. In the context of ERS, a pragmatic methodological approach suits examining the real-world environment to produce results that are relevant to stakeholders and the research community whilst being rigorously sound [26]. 

First, three authors familiarised themselves with the data via reading transcripts and listening to audio recordings. Second, an initial set of codes were set either by identifying recurring words within the dataset or words of interest to the authors, for example, “motivation”, “encouragement”, “communication”, and “autonomy”. Generated codes were cross-referenced against each other and, where appropriate, combined. For example, “welcoming”, “friendly”, and “comforting” were grouped together to create the code “Interpersonal support”. Within this step, codes were worded appropriately to allow contextual understanding of the code. For example, “motivation” became “provide motivation”; thus, it became clear that providing motivation came from the instructor. Furthermore, some codes may appear similar; however, they have different contextual meanings. For example, Code 1 (participants’ personal goals) is in reference to what the participants have described to the instructor about what they want to achieve, and we give the example of weight loss. This code is positioned within the theme “instructor’s perception of participants views”. Code 35 (goal setting), which is positioned within theme “instructors’ use of different techniques which could help elicit behaviour change”, is in relation to a technique used by an instructor. Thus, these are different but closely interlinked, as the instructor will use goal-setting as a technique to help a participant achieve their personal goals. All generated codes related to uptake and attendance of ERS and adherence to the prescribed exercise. Transcripts were coded by hand, constantly revisited and cross-referenced, throughout this first iterative step by three authors (Colin B. Shore, Trish Gorely, Gill Hubbard). 

The third step allowed for authors to search for initial themes. Each initial numbered code (see Table 1) was grouped with similar codes to form descriptive themes. Coded data were grouped into four descriptive themes: (i) the role that instructors perceive they have and approaches those instructors take to motivate the participants to take up, attend exercise referral and adhere to their exercise prescription; (ii) instructors’ use of different techniques, which could help elicit behaviour change; (iii) instructors’ perceptions of participants’ views of exercise referral schemes; and (iv) barriers towards providing an ERS. Steps four and five allowed authors to revisit, check and refine themes. Throughout all stages described above, draft analyses were circulated between three authors (Colin B. Shore, Trish Gorely, Gill Hubbard). Face-to-face meetings allowed discussions about initial coding and descriptive themes, thereby reaching a consensus on descriptive thematic analysis and interpretation. 

## 3. Results

One hundred per cent of exercise referral instructors employed at the time of the study agreed to participate. In total, six interviews with a median duration of 44 (37–53) min were conducted. Fifty per cent of instructors were female, with a median of 3 (1–12) years of experience working as an exercise referral instructor. One instructor, interviewed as a study participant, was undergoing level three referral qualification. They were still included within the study as they delivered the programme in conjunction with a qualified instructor. Thirty-five initial codes were created. These codes were grouped into four descriptive themes. The following Results section is organised to represent the four descriptive themes with the most representative quotations from participants.

### 3.1. Role That Instructors Perceive They Have and Approaches They Take to Motivate the Participants to Take Up, Attend Exercise Referral and Adhere to Their Exercise Prescription 

Instructors highlighted that a key role of theirs was to provide clear information about what the ERS entails. This role was twofold and acted as an opportunity to sell the benefit of the scheme and was key to tackling the lack of information provided by the HCP who made the referral. In describing the purpose of the scheme and how exercise and PA were beneficial to many long-term conditions, instructors were selling the idea of lifestyle improvement for participants. For instance, the technique of shaping knowledge was used by instructors to sell the benefits that participants may anticipate experiencing, e.g., improvement to activities of daily living, such as not being out of breath when carrying shopping bags up the stairs. Furthermore, instructors saw their role as painting a picture that exercise is not to be feared and can be enjoyable. Evidentially, the first session with participants is very important. However, it is not known the extent to which instructors are trained to take on these roles or if their techniques are based purely on their lived experience.


*“I see that as part of my job, sit the client down at the point of consultation to say, right, well this is what we’re gonna do with you, and this is what you’ll hopefully see the benefits is”.*
(In2)


*“I try and give them the reason behind why we give them stuff, so I like to let them actively know, this is why we’re doing this I want them to know the reason why we do things, it is important to do”.*
(In5)

This study did not set out to measure attendance; however, all instructors acknowledged that low attendance or dropping out of ERS is high. The biggest challenge that instructors encountered in engaging participants to uptake or in keeping participants in the programme is participant motivation. Communication was perceived as key to building a strong rapport between themselves and participants, which, in turn, was crucial to encouraging attendance and adherence. 

Instructors believed that it is vital that participants have a good experience the first time that they attend since this will encourage regular attendance. This included making the participant feel at ease, connecting with the participant around their goals, answering any questions participants had, dismissing any anxieties held by participants. For example, demonstrating that attendees of leisure centres come from many walks of life, and they (participants) are just as welcome there as anyone else. Another approach described was that of social support. Instructors would make sure that the participant did not feel that they were on his or her own but that they (instructor) were there for that participant. Instructors believed that if participants left feeling comfortable and that their needs were understood, there was a strong chance they would return and become regular attendees. 


*“Getting people into the gym, I think if they get that good experience especially in the first time, if they have a good first impression (…) Keeping people motivated the hardest part…if you interact with them, even if it’s just a case of… but two or three times, just ‘til they get comfortable with it, but you’re touching base to say how it’s feeling?”.*
(In5)

Instructors describe another of their roles was to promote and motivate long-term engagement with exercise after the ERS ends. This was irrespective of the participant obtaining a leisure centre membership. However, there was a business need for the leisure centre to make a profit. Upon asking if there was external pressure to keep participants attending and coming back to the leisure centre, there was a mixed response. Those instructors who expressed external pressures conceptualised it from a business perspective but ultimately felt that if participants performed more exercise, that was all that mattered. 


*“I mean, obviously we’re (leisure centre as a business) in to make money but people that actually adhere to the programme and then not necessarily taking a membership but staying active. Maybe not through us. I want to get people, as I said, to maintain what they’re doing or actually increase what they’re doing.”.*
(In4)

### 3.2. Instructors’ Use of Different Techniques Which Could Help Elicit Behaviour Change 

Instructors declared that there was no formal behaviour change theory embedded within the ERS that was intended to improve uptake, attendance and adherence. However, as illustrated by the quotations, instructors appeared to make use of a variety of common behaviour-change techniques (BCTs) in order to improve attendance and adherence; for instance, problem-solving, self-monitoring, social support (practical and emotional) and identification of self [22]. In some cases, the use of BCTs was targeted; in others, it was more implicit. Furthermore, instructors did not distinguish between using BCTs to increase attendance (i.e., regularly attending the bi-weekly sessions) and adherence (i.e., performing and completing the prescribed exercise within a referral session); rather, they were packaged as a whole. Different BCTs were used by different instructors, and use was based on the experience of the instructor and the relationship they had with the participant. In5 used self-monitoring by encouraging participants to reflect on their previous exercise ability so that they could see how far they had progressed; In3 used verbal encouragement and distraction. In2 and In6 used social support to achieve different outcomes. In2 used social support by encouraging participants to give each other lifts in cars to attend ERS. In6 used social support to group people together so that they could support one another to engage and adhere to the exercise prescription. 


*“we just try and push them past the barriers and just try and say, look, where did you come from, if you can think back when you started phase three, for instance”.*
(In5)


*“I think I probably have a bit of a jokey, kind of, sense with them and say, come on, let’s get this done, let’s get that done and egg them om (…) just stay really upbeat with them”.*
(In3)


*“there’s almost a social side of the class, as well (…) if they’re in that group environment, they’ll know that their pal, Jessie, is coming in with them (…) couple of occasions where they’re car-pooling (…) because they’re coming in at the same time as Billy, and they can have a chat about whatever, so yeah, so that side of it as well that’s a good retention”.*
(In2)


*“try and maybe group a couple together with the same (…) I think it kind of gets them talking, gets them to open up a bit more, and I think it gets them to motivate one another as well”.*
(In6)

### 3.3. Instructors’ Perceptions of Participants’ Views of Exercise Referral Schemes

Instructors expressed the importance of creating an environment that is welcoming and appropriate for the participant, alleviating any fears they (participants) might have about exercising or being judged. Instructors described how participants expressed emotions about ERS, such as being scared or anxious, which they perceived was likely to influence attendance and adherence. Therefore, the role of the instructor becomes more about mentoring the participant and providing reassurance and guidance to help ease concerns and trying to educate participants on the benefits of continued attendance and adherence to the exercise programme. 


*“Some of the clients know they’re coming into the gym, and they’re like, oh, I don’t really want to do this. We also see that about, when we take their blood pressure, their heart rate is through the roof before they even come in, it’s like 100 plus, and I’m like, right, are you a bit nervous? and they are nervous. Because they’re coming into an environment, they’re unaware of, they don’t know what it’s all about, they might have a misconception in their head in regard to what a gym is”.*
(In2)

### 3.4. Barriers towards Providing an Exercise Referral Scheme

The physical infrastructure (e.g., size or layout of the gymnasium) of the ERS setting and the timings of ERS sessions were perceived as a barrier to encouraging attendance and adherence. Across all four leisure centres sites, the gymnasium where ERS participants exercised was open to the general public, which could be off-putting, according to instructors; for example, loud noises, as more experienced gymnasium users might drop heavy free weights. Furthermore, because the gymnasium, in some cases, was small, the physical limit on capacity might lead to a situation where there could be a lack of equipment available, meaning a participant might have to use an item of kit they are unfamiliar with. Instructors described that participants new to exercise could easily be put off in such circumstances, and this will influence their attendance. Instructors expressed that extending the length of their ERS has been a positive step to help increase attendance (this was a change implemented prior to this study being conducted, where the ERS used to be 8 weeks and is now 12 weeks). This additional support was key to allowing participants to establish exercise habits over a longer period, rather than participants being transitioned through the system at pace. 


*“I think the people not coming back is more to do with the environment more than the session, they all say, was great, thank you, loved it, come in one time and it’s not very quiet, people dropping weights and things like that and you can see their heads starting to go and it’s out of my control unfortunately”.*
(In4)


*“it would be nice to get more participants in, but I think you would need to disperse it over the whole week rather than having them in the two-hour slots and the two days”.*
(In6)


*“Well, we did make a change there (…) it used to only be an eight-week block, so we increased that to 12 just to try and improve the adherence of the participants, so I think that’s been one good improvement”.*
(In6)

## 4. Discussion

### 4.1. Main Findings

The present study describes several processes of how exercise referral instructors facilitate uptake, attendance and adherence to prescribed exercise. The role of the exercise referral instructor is multifaceted, including that of “sales pitch”, education, problem-solver and motivator. Instructors make use of a handful of common BCTs in order to elicit attendance at ERS and adherence to the prescribed exercise. 

Previous research has shown that exercise instructors said that they were successful in engaging older adults in exercise classes if participants were part of a specific referral system [27]. However, this manuscript, consistent with previous work [10], suggests that HCPs do not possess the relevant information about the scheme, or if they do, according to the perceptions of exercise referral instructors, communication to the participant about ERS is typically lacking. This suggests that referred participants do not know what they are going to be engaging with. This means the initial role of the instructor is that of an information provider. Considering this, the present study’s findings are consistent with previous studies regarding the intimidating nature of leisure centre environments and anxiety held by participants towards exercising [14,28]; a lack of communication to the participant may not help ease any anxiety towards exercising. Therefore, a key role of exercise referral instructors is to normalise and sell exercise as a behaviour. By communicating with participants and forging a positive relationship with participants, instructors help to provide a strong positive impression of ERS, reducing any fears. The outcome of this is a motivated participant who uptakes ERS and regularly attends. Participants who received support from and perceived instructors to be supportive have reported higher identified motivation to uptake ERS [29]. Previous work has shown that exercise referral instructors demonstrated a strong sense of professional affiliation, in which they felt "ultimately responsible" for ERS delivery [10]. This is evident in the present study, where instructors were committed to motivating participants to attend and adhere to the exercise prescription and benefit from ERS, even if they did not take a membership with the leisure centre. 

The evidence presented points to instructors providing high levels of support and communication at the start of the programme to help foster attendance and adherence to the prescribed exercise. This is consistent with previous research [16,30] demonstrating that providing social interactions are an effective behaviour-change approach to help maintain attendance in the programme [17,18]. Furthermore, social support has been cited as having an important impact on older adults’ intention to participate in exercise [31]. Additionally, it has been demonstrated that instructors who have undergone motivation training were able to engage participants in higher attendance levels of exercise classes [32]. There is promising evidence demonstrating the beneficial impact of motivational interviewing across a number of different outcomes, settings and patient groups. These include, but are not limited to, aiding weight loss and adherence to healthy lifestyles [33], community-dwelling older adults [34], and certain patient groups, such as people with type-2-diabetes [35] and stroke [36]. However, the fidelity and proficiency of delivering motivational interviewing have been cited as key factors for success [35], while cost-effectiveness is somewhat unclear [37,38]. Moreover, one study which tried to introduce motivational interviewing into ERS in Wales found some resistance from instructors and a number of difficulties combining motivational interviewing with other activities [39]. Our study is unable to provide further details as to whether the exercise referral instructors had received additional training. It is, therefore, a recommendation that this should be an avenue of further research in order to further establish the efficacy of motivational interviewing within ERS. 

Instructors described offering practical guidance on supporting participants to exercise. Consistent with previous findings [28], practical support was inseparable from the discussion of the need to provide interpersonal support to build confidence and to motivate participants to attend and adhere to the programme. Moreover, such support was often used as a tool to help distract participants from, for example, worrying about how long they had been left on the treadmill, to help aid adherence to the prescription. Distractions have been shown to increase tolerance of high-intensity exercise, suggesting that a change in attentional processing from internal (physical sensations) to external perspective (distractions) may have facilitated this improvement [40].

Henderson [10] described service provision and the strategic management of ERS to be at odds with each other. The present study’s findings appear to reflect such tensions. Instructors described two key strategic factors, out of their control, which they perceived to have a negative impact on attendance: size and sole use of the gymnasium and timings of the programme. Previous qualitative research reported that participants were more likely to exit a scheme early due to a lack of ERS staff availability, support and timings of ERS [14]. Instructors in the present study described that sessions ran during a quieter period of the day; however, the gymnasium where participants exercised was also open to the fee-paying public. If physical space was at a premium, alongside non-ERS exercisers creating noise, or the exercise referral instructor had to divert attention away from ERS participants, this would often put participants off from attending, as they were not receiving the focused attention that was anticipated. Additionally, instructors felt that participants were limited by programme timings and wished they could offer more support and more opportunities to attend. 

This study did not set out to test if certain factors were having a greater impact on uptake, attendance or adherence; rather, the objective was to describe the current practice to ensure uptake, attendance and adherence. Exercise referral instructors deliver ERS classes in addition to their other roles within the leisure facility, including within a programme of classes. Therefore, it seems that the greatest opportunity for ERS instructors to have an impact on participants attendance would be to afford them more time with participants and facilities that allowed staged integration, so as not to deter those less active from exercising with regular members of the public, and offer a greater choice of times to attend. Currently, ERS does not facilitate such a service, which may influence uptake and sustained attendance. This may, in turn, impact ERS’s ability to play an effective role in primary and secondary prevention and self-management of NCDs. 

### 4.2. Strengths and Limitations

The debate often surrounds qualitative research and the confidence we can place upon conclusions, and subsequently, the ability to extrapolate or generalise to the population. This often manifests itself via sample sizes and “how many interviews is enough” to declare confidence in the findings [41]. In the current study, one hundred per cent of ERS instructors employed within the scheme participated in the study. Therefore, a strength of the study includes a 100% response rate of exercise referral instructors employed at the sites participating in the study. Subsequently, views expressed reflect those of all the exercise referral instructors, not just a proportion. However, the ERS studied was in a rural location and was comparatively small, with limited facilities. This suggests that some findings may not translate to larger ERSs, which may have more facilities and opportunities. 

A strength of the study is the rigour undertaken in the reporting of the research process [42,43]. That is, the credibility of the findings is evidenced through the accurate and truthful depiction of exercise referral instructors’ answers and perceptions about their experiences of ERS. Dependability of analysis was achieved through triangulation between three authors (Colin B. Shore, Trish Gorely, Gill Hubbard) and reviewing transcribed material at different stages to validate findings. 

## 5. Conclusions

Perceptions from exercise referral instructors are of a communication gap between the referring HCPs, participants and the ERS. This lack of information is a barrier to ensuring effective uptake and subsequent attendance. This presents an opportunity to consider research to explore a role for exercise specialists to act as the go-between from general practice and health services to ERS (e.g., link practitioners). Exercise referral instructors inhabit many different roles, from salesperson, educator, problem-solver, and mentor to confidant in order to facilitate attendance or adherence. Instructors utilise a handful of BCT, some of which are targeted, while others are unconsciously used and dependent on their own personal style and the relationship they have with the participant. However, little is known to what extent additional training and support (e.g., motivational interviewing) are given to exercise referral instructors and what influence this may have on participants. It is therefore recommended that future research considers the need for ongoing education and peer support for instructors. Without a meaningful understanding of what current approaches are in practice, ERS may continue to face criticism over its effectiveness. However, our findings provide perspectives that may help us to further understand how health and leisure services, researcher and co-production with participants can design successful ERS.

## Figures and Tables

**Table 1 ijerph-19-00203-t001:** Coding and descriptive themes of exercise referral instructor’s perspective of motivating and supporting uptake attendance and adherence to exercise referral schemes.

Code Number	Code	Theme 1 ^a^	Theme 2 ^b^	Theme 3 ^c^	Theme 4 ^d^
1	Participants personal goals (e.g., weight loss)			✓	
2	Creating personalised, realistic and achievable prescriptions and targets	✓			
3	Provide motivation		✓		
4	Provide interpersonal support (e.g., welcoming/friendly/encouragement/reassurance)	✓	✓		
5	Participant finds gym intimidating			✓	✓
6	Referring health professionals lack knowledge about the scheme				✓
7	Provide gold standard customer service		✓		
8	Creating a positive first impression	✓			
9	Provide appropriate level of care for the condition (e.g., cardiac patients)	✓			
10	Encourage autonomy and independence of participant	✓	✓		
11	Instilling a foundation to create lifelong habits. Building blocks	✓			
12	Participant-centred	✓			
13	Provide feedback (e.g., showing progression)		✓	✓	
14	Poor communication from healthcare professional to participant				✓
15	Time of session tricky for some participants			✓	✓
16	Providing knowledge and benefits of becoming active	✓	✓		
17	Restrictive gym size				✓
18	Being able to manage people through the programme (e.g., reining people back in)		✓		
19	Instructors are frustrated when patients drop out				✓
20	Instilling a level of education for participants	✓	✓		
21	Clinical improvements as a reinforcement tool		✓		
22	Instructors need to have problem-solving skills	✓			
23	Building relationships with participants	✓	✓		
24	Participants express fear of exercising			✓	
25	Instructors want to create replicable exercises of activities of daily living	✓			
26	Instructors justify why they prescribed an exercise	✓			
27	Instructors create variety to keep people motivated	✓	✓		
28	Referral system creates bottleneck, which can hamper service delivery				✓
29	Participants view exercise as too much hard work			✓	
30	Participants have an epiphany, a realisation that exercise is positive		✓		
31	Participants lack knowledge about the scheme			✓	✓
32	Instructors try to group people to create friendship, show they are not alone		✓		
33	Lack of equipment				✓
34	External distractions (e.g., having to run on gym floor at same time)				✓
35	Goal-setting		✓		

^a^ Role that instructors perceive they have and approaches they take to motivate the participants to take up, attend exercise referral and adhere to their exercise prescription. ^b^ Instructors’ use of different techniques that could help elicit behaviour change. ^c^ Instructors’ perceptions of participants’ views of exercise referral schemes. ^d^ Barriers towards providing exercise referral scheme.

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
