# Peer review of "Exercise Referral Instructors’ Perspectives on Supporting and Motivating Participants to Uptake, Attend and Adhere to Exercise Prescription: A Qualitative Study"

_ijerph, 2021, doi:10.3390/ijerph19010203_

Round 1
Reviewer 1 Report
General comments
The authors conducted a qualitative study evaluating the perception of the exercise instructors about the exercise referral schemes in Scotland. The authors explored the perceptions of the exercise referral scheme in 6 exercise referral instructors using semi-structured interviews. They analysed the verbatim and found four main themes. The topic is relevant, and the quality of the study is good. However, the current article has some drawbacks (e.g. the four themes) which should be fixed before publication.
The main issue of this article is the four themes (Page 4, lines 173-180). The 35 codes are listed in the Appendix 1A (or A). The codes are well defined. What is the difference between codes 1 and 35? however the themes could be different. The 1st theme could be divided in two separate themes: “role that instructors perceive they have” and “approaches they take to motivate the participants”. However, the latter overlaps the theme 2. Currently, there is an evident overlap between themes 1 and 2 (and also between themes 3 and 4). Indeed, 6 codes (4, 10, 16, 20, 23, 27) are used both in themes 1 and 2. I am wondering if these 6 codes (and possibly other codes) could be isolated to create a new theme about exercise literacy. Moreover, 3 codes are used both in themes (5, 15, 31). I am wondering if it would be possible to create themes without using the same codes in different themes. It will contribute to better discriminate the themes. Themes could also be changed into the following sections:
- Items that could be changed or controlled by the instructors
- Items that could be changed or controlled by the patients
- Items that could be changed or controlled by the owner of the leisure centre
- Items that could be changed or controlled by the healthcare professionals who referred the patients
- Items that could not be changed or controlled
In addition, it will be easier for the readers to have the full description of the codes in the description of the themes. The appendix A could be replaced by a table in the text.
Specific comments
Abstract
Page 1, lines 12-13. “To” is missing between disease and increase
Page 1, lines 13-16. The sentence is too long and should be rewritten (exercises referral schemes is repeated twice in the sentence”
Page 1, lines 16-20. The themes may be rewritten (please refer to the general comments).
Keywords:
Page 1, lines25-26. What are the numbers after each keyword?
Introduction
Page 2, lines 44-46. Please define in the introduction “uptake”, “attendance” and “adherence”.
Page 2, lines 46-48. What are the other factors explaining the low uptake, attendance, and adherence?
Page 2, lines 61-62. Is there a role model effect of the instructors on the participants? Please add references if any.
Page 2, lines 65-67. What is the percentage of patients using the “exercise prescription”? This percentage may be related to the motivation skills of the health care professionals who referred the patients. Other factors may also explain this percentage (e.g. stage of behavioural change of the patients, etc.) What is the percentage of patients who are attending the first exercise session? What is the attrition rate? What is the level of attendance? These data could help the readers to understand the scale of the problem.
Page 2, lines 69-70. Please check the sentence.
Materials and methods
Page 2, line 88. Is it expensive (or affordable) for the patients?
Page 2, line 89. It is renewable? What is happening after the end of the 12 weeks? Please describe here (it was described in the discussion, but it is probably better to describe here what is happening at the end of the programme).
Page 2, line 89. Please describe how the patients are contacting their local leisure centre? By phone? On site? How do the patients know where to go (Is the contact of the leisure centre(s) provided within the “prescription”?)?
Page 2, line 92. Please add “initial appointment” in this sentence. I am wondering if the patients could also take part to an exercise session at the initial appointment.
Page 3. Lines 97-102. It is not clear if there are group classes.
Page 3, lines 102-107. The definitions should be in the introduction.
Page 3, line 110. Please define “GP”.
Page 3, lines 111-113. These two sentences should be in the results section.
Page 3, line 129. Please move the characteristics of the recordings (i.e.Olympus VN-731PC) in the next paragraph.
Page 3, lines 129-133. Please move these two sentences in the results section. Please add a table to describe more the participants: gender, age, experience, qualification (exercise, etc.).
Page 4, lines 146-148. What are the questions? Please add a list in the appendix.
Page 4, lines 161-184. Please return to the line for each step.
Page 4, lines 173-179. The four themes may be modified. Please refer to the general comment.
Results
Page 5, line 196. Please delete the coma after “knowledge”.
Pages 4-5, lines 191-206. It seems that the first meeting with the patient is essential. Are the instructors trained for this first meeting? What can they do to improve the first meeting?
Page 5, lines 212-225. How can the instructors improve the first exercise session? What is the drop-out rate after the first exercise session?
Page 5, lines 228-229. I am wondering if the business need for the leisure centre to make a profit is a major issue of the ERS. What is the price per session after the end of the ERS? It is affordable for the patients? Do the instructors have benefits if patients are taking a leisure centre membership? Are the instructors paid by the leisure centre (or are the instructors “renting” the leisure centre and make profit individually with their own clients)? What is the motivation of the instructors to participate to the ERS? It may be utopic to believe that patients will remain active by their own after the end of the ERS. It seems unlikely that exercise literacy will be enhanced within 12 weeks.
Pages 5-6, lines 237-267. Is it really “BCTs” or only components of the BCTs? BCTs are probably not used by the instructors. Therefore, I am not sure if the use of the wording “BCT” is appropriate. Moreover, the authors do not know if the instructors were trained for some BCTs. There are only 6 instructors it may be feasible to ask them again.
Page 6, lines 269-271. Creating a welcoming environment is a consequence. It is not a perception of the participants’ views.
Page 6, line 277. I am wondering if all instructors used the word “clients” for the participants who are actually patients. It may be more presented and discussed.
Page 6, line 284. What is “physical infrastructure”?
Page 6, line 287. “which could be off-putting”: is it according to the patients, the instructors, or the authors?
Page 6, line 290. “lack of equipment available”: is it only related to the size of the gymnasium and/or to the use of the gym by the general public?
Page 6, line 293. It should be stated in the manuscript that the ERS was extended from 8 to 12 weeks. Who decided to extend the ERS?
Page 7, lines 297-300. Why there is no dedicated slots for patients with ERS and patients who want to continue?
Page 7, line 306. Instructor 1 was never quoted.
Discussion
Page 7, lines 342-344. There are only 6 instructors it may be feasible to ask them again.
Pages 7-8, lines 346-354. I do not understand why the authors are discussing the distraction (external focus of attention). It was not introduced and presented earlier in the manuscript.
Page 8, lines 369-371. This sentence may be moved to the limitations section.
Page 8, line 386. “100%” may mislead the readers. They are “only” 6 participants in the study.
Pages 8-9, lines 397-414. The conclusion is too long.
Page 9, line 402. ERS with “connectors” were tested in the Netherlands. Please find some references in the following article (in the section concerning “Exercise referral schemes with connectors”): https://academic.oup.com/heapro/article/34/4/877/5035043.
Appendix
Page 10, line 431. Please see the general comments.
Page 11, line 434. The appendix 1B (or B) was not cited in the text. These terms should be defined in the introduction of the article. Therefore, this appendix could be deleted.
References
Page 11, line 463. Title should be in italic. Please be consistent throughout the reference list.
Page 11, line 464. The DOI has no hyperlink. Please be consistent throughout the reference list.
Page 11, line 466. Journal title is abbreviated. Please be consistent throughout the reference list.
Page 12, line 517. The issue (4 for the present reference) is not indicated. Please be consistent throughout the reference list.
Reviewer 2 Report
After reviewing the manuscript, I do not see much merit or novelty in your findings. You have no actual hypothesis and this is a purely descriptive manuscript with no actual data but just a series of quotes. This would be better served as an abbreviated commentary in an exercise behavior journal.
Major.
Introduction:
You should define NCD. Also, not just physical inactivity can lead to the development of NCD. You should at least mention those other risk factors.
Line 30 - The paper you cited lower respiratory infection is the 4th leading cause of death this sentence and reference is very unclear.
It is unclear to me if the instructor is the one soliciting the individual for exercise or instructing the actual exercise.
What is your hypothesis? I understand you have an aim but it is unclear what your actual question is.
Methods:
Line 85 – Define PA guidelines. What are they specifically?
Six interviews seems to be a very low number.
What are the qualifications?
Results:
This section contains no data. Just quotations. When reading the journals description of what would fall into this category I do not thing this fits.
Discussion
You are basing your discussion off of a conversation with 6 exercise instructors of some sort. I am not sure what the objective of point of this paper was. It is not data drive in the very least.
Minor
Line 48 – Very awkward sentence and needs to be rewritten
Lines 129-133 may be better in results
Reviewer 3 Report
ABSTRACT
This section is well written, presents the research purpose adequately and briefly summarises the results achieved as well as a preview of the main conclusion.
Introduction
Who decides that a person is inactive, what levels of inactivity classify him/her as an inactive person? Please include references to support this statement.
Lines 40-43. Excellent references to support the arguments.
Overall this section allows the future reader to adequately approach the state of the art regarding ERS.
Materials and Methods
2.1. Study Design.
Ok
2.2. Exercise Referral Scheme.
Line 81-82. Please avoid breaking syllables between words... "dif-ferent"... Review it throughout the text.
The two weekly sessions with instructor do not indicate the duration of each session: 1 hour, 2, hours... 5 hours?
Nor is there any indication of the nature of the sessions, or the intensity control ... distance and time, accelerometry, beats per minute, subjective perception of fatigue ...?
Nor is it indicated how many students each instructor has.
All these elements must be taken into account in this methodology section as the results depend to a large extent on them.
Recruitment and participant inclusion.
Very well explained.
Qualitative data collection.
This section describes well how the data collection has been done and even includes a pilot test.
2.5. Qualitative analysis
Very well explained.
3. Results
In this section the four descriptive themes are well summarised.
The examples are clear and help the reader to understand the research findings.
Discussion / Conclusions
These are undoubtedly the two best sections of the article with a huge potential for future citation.
They present well, in the discussion, comparisons with previous research and clearly point out new findings.
In the conclusion section they point out the possibilities for a better performance of these professionals for the future, without forgetting the shortcomings detected, especially in the communication with their trainees.
Round 2
Reviewer 1 Report
Comments to the authors
The authors’ responses to my comments and the related modifications of the manuscript were mainly satisfying. I have few minor remarks.
Page 3, line 112. Please replace the dot by a comma.
Page 8, line 393. The effectiveness of the motivational interviewing could be discussed. The effectiveness of motivational interviewing may be plausible in a research setting. However, it would work only in specific circumstance in the “real word”. In addition, the cost-effectiveness of the motivational interviewing could also be discussed
Page 11, figure 1. The figure 1 should be modified in table 1. The table should be as follow
- 1st column: code number (from 1 to 35)
- 2nd column: code description
- 3rd column: theme 1 (yes/no or a colour code to know if the code has been used or not in the theme)
- 4th column: theme 2 (yes/no or a colour code to know if the code has been used or not in the theme)
- 5th column: theme 3 (yes/no or a colour code to know if the code has been used or not in the theme)
- 5th column: theme 4 (yes/no or a colour code to know if the code has been used or not in the theme)
Themes should be defined in the column header or in the table footnote.
Page 13, line 545. Please check ref 19: the journal title is abbreviated.
Reviewer 2 Report
The paper is much improved. The introduction is more thorough and the methods are more clear. I still feel as though it study has very little impact and would be of low interest to readers. There are no measurable outcomes. I am not sure what value this would bring to the scientific community. The number of individuals you interviewed is low. Although the data is qualitative to could be better organized.
